# Antibacterial activity of eravacycline against *Klebsiella pneumoniae* isolates: an *in vitro* study

Yuanzhi Gong,[1] Yuhao Liu,[1] Yunlou Zhu,[1] Shikui Mu,[1] Hanlu Gao,[1] Xin Jing,[1] Yingying Du,[2] Sheng Wang[2]

**ABSTRACT**   Eravacycline, a novel tetracycline antibiotic, may be an effective treatment option for *Klebsiella pneumoniae* infections. We thus conducted an *in vitro* susceptibility analysis for eravacycline in 211 *K. pneumoniae* isolates. Eravacycline achieved an overall susceptibility rate of 86% against *K. pneumoniae*, and the sensitivity rates to carbapenem-susceptible *Klebsiella pneumoniae* and carbapenem-resistant *Klebsiella pneumoniae* (CRKP) strains were 100% and 84%, respectively. The combined drug sensitivity test *in vitro* for eravacycline and polymyxin B demonstrated a synergistic effect in 20% of eravacycline-resistant CRKP strains.

**IMPORTANCE**  Carbapenem-resistant *Klebsiella pneumoniae* (CRKP) is a global priority pathogen due to its limited therapeutic options and high morbidity, which urgently needs new antibiotics that are not affected by common resistance mechanisms. As the next generation of fluorocycline antibiotics, eravacycline can circumvent the tetracycline-specific resistance pathways by its structural modifications, which may be a candidate antibiotic for CRKP infections. The present study evaluated the *in vitro* activity of eravacycline against CRKP strains, including carbapenemase producers, and explored synergistic interactions with existing antibiotics. By determining the efficacy spectrum and combinatorial potential for eravacycline, this study will directly guide clinical strategies to combat infections caused by CRKP and optimize treatment regimens for high-risk populations.

**KEYWORDS**   *Klebsiella pneumoniae*, eravacycline, polymyxin B, antibiotic combination treatment

*K*lebsiella pneumoniae, a member of the Enterobacterales order, is one of the most common nosocomial pathogens (1). In particular, carbapenem-resistant *Klebsiella pneumoniae* (CRKP) is a significant pathogen within hospital settings, leading to serious infections characterized by high morbidity and mortality rates. In 2019, the Centers for Disease Control and Prevention identified CRKP as one of the most critical "urgent public health threats," underscoring the necessity for immediate intervention strategies (2).

The primary mechanism facilitating carbapenem resistance in *K. pneumoniae* is the production of carbapenemase, notably KPC carbapenemase and metallo-β-lactamases (3). In recent years, therapeutic options for combating carbapenem-resistant Enterobacterales infections have largely been confined to regimens incorporating polymyxins and tigecycline (4). Notably, the clinical efficacy of conventional antimicrobial therapies is steadily decreasing, a trend exacerbated by the rapidly accelerating emergence of multidrug-resistant pathogens (5). These challenges highlight the critical imperative to prioritize the development of novel antimicrobial agents and therapeutic strategies to combat the escalating global burden of antimicrobial resistance.

Eravacycline, a fluorocycline antibiotic synthesized from the tetracycline core structure with alterations in the D ring, exhibits a wide spectrum of activity against

**Editor** Andrea M. Prinzi, bioMerieux Inc., Denver, Colorado, USA

**Peer Reviewers** Benno H. Ter Kuile, Universiteit van Amsterdam, Amsterdam, the Netherlands; Mor N. Lurie-Weinberger, Tel Aviv Sourasky Medical Center, Tel Aviv, Israel

Address correspondence to Yingying Du, 2211696@tongji.edu.cn, or Sheng Wang, wangsheng@tongji.edu.cn.

Yuanzhi Gong and Yuhao Liu contributed equally to this article. The order of co-first authors was determined based on their relative contributions.

The authors declare no conflict of interest.

See the funding table on p. 7.

Gram-positive and Gram-negative bacteria and anaerobes (6). In 2014, the U.S. Food and Drug Administration approved intravenous eravacycline for the treatment of complicated intra-abdominal infections in adult patients, which were attributed to various enteric microorganisms (7, 8). Eravacycline targets the 30S ribosomal subunit, which impedes bacterial protein synthesis. This mechanism of action ensures its efficacy, particularly against bacterial strains resistant to the combined effects of β-lactam and β-lactamase inhibitors (9). *In vitro* studies have shown that eravacycline is highly stable in the presence of serine- and metallo-β-lactamases (10). This stability indicates that eravacycline may be effective against infections caused by CRKP.

However, the performance of eravacycline against *K. pneumoniae* has not yet been documented sufficiently (11). The susceptibility of eravacycline against CRKP in combination with other antimicrobial agents has not been investigated. Therefore, we conducted an *in vitro* analysis of eravacycline activity using 211 clinical *K. pneumoniae* isolates and assessed combinatorial regimens to identify synergistic strategies against CRKP.

## RESULTS

### Resistance genes in *K. pneumoniae* isolates

Between 2021 and 2023, 211 *K. pneumoniae* isolates were collected from Shanghai Tenth People's Hospital, Tongji University. Of these, 183 carried carbapenemase, including 156 with KPC carbapenemase and 17 with metallo-β-lactamases, and 10 isolates co-harbored both KPC carbapenemase and metallo-β-lactamases. The remaining isolates were carbapenemase-negative and exhibited *in vitro* susceptibility to carbapenems.

### *In vitro* susceptibility testing

We investigated the susceptibility of 211 strains of *K. pneumoniae* to eravacycline (Fig. 1). Among the 211 tested strains, 86% were susceptible to eravacycline. Of the 183 CRKP strains, 84% (153/183) were susceptible to eravacycline, while all the remaining 28 carbapenem-susceptible *Klebsiella pneumoniae* (CSKP) strains were sensitive to eravacycline. We subsequently categorized the strains according to their different enzyme types and analyzed their minimum inhibitory concentration (MIC) against eravacycline (Table 1). The MIC distributions are presented in Fig. 1, and the *in vitro* susceptibility of the CRKP strains to eravacycline showed no significant differences among enzyme classes: susceptibility rates were 84% for KPC-2 producers, 77% for metallo-β-lactamase (MBL) producers, and 90% for dual carbapenemase (class A + B) producers. These results indicate that the bactericidal activity of eravacycline is not substantially impacted by the type of carbapenemases (KPC-carbapenemase, metallo-β-lactamase, or co-producers) under *in vitro* conditions.

The MIC results showed that tigecycline has an 87% susceptibility rate against CRKP, with MIC50 and MIC90 values of 2 and 4 mg/L, respectively. The MIC values and sensitivity rates for both tigecycline and eravacycline are summarized in Tables 2 and 3. The *in vitro* susceptibility of CRKP isolates to polymyxin B, meropenem, amikacin, and fosfomycin was evaluated. Notably, 30 eravacycline-resistant CRKP isolates exhibited complete phenotypic resistance to meropenem (MIC range: 64–256 mg/L) and fosfomycin (MIC range: 64–256 mg/L), whereas susceptibility to amikacin varied (MIC range: 2–8 mg/L). Polymyxin B demonstrated the highest susceptibility, with all eravacycline-resistant strains exhibiting a 90% susceptibility rate (MIC range: 0.125–16 mg/L). The MIC distributions and resistance rates are summarized in Table 4.

### Checkerboard test

The observed suboptimal efficacy of eravacycline against CRKP prompted us to seek more effective antibacterial treatment approaches. Thus, we examined the efficacy of combination therapies against eravacycline-resistant CRKP strains. Synergistic effects were found in 20% of eravacycline-resistant CRKP strains (KP-569, KP-572, KP-530, KP-638,

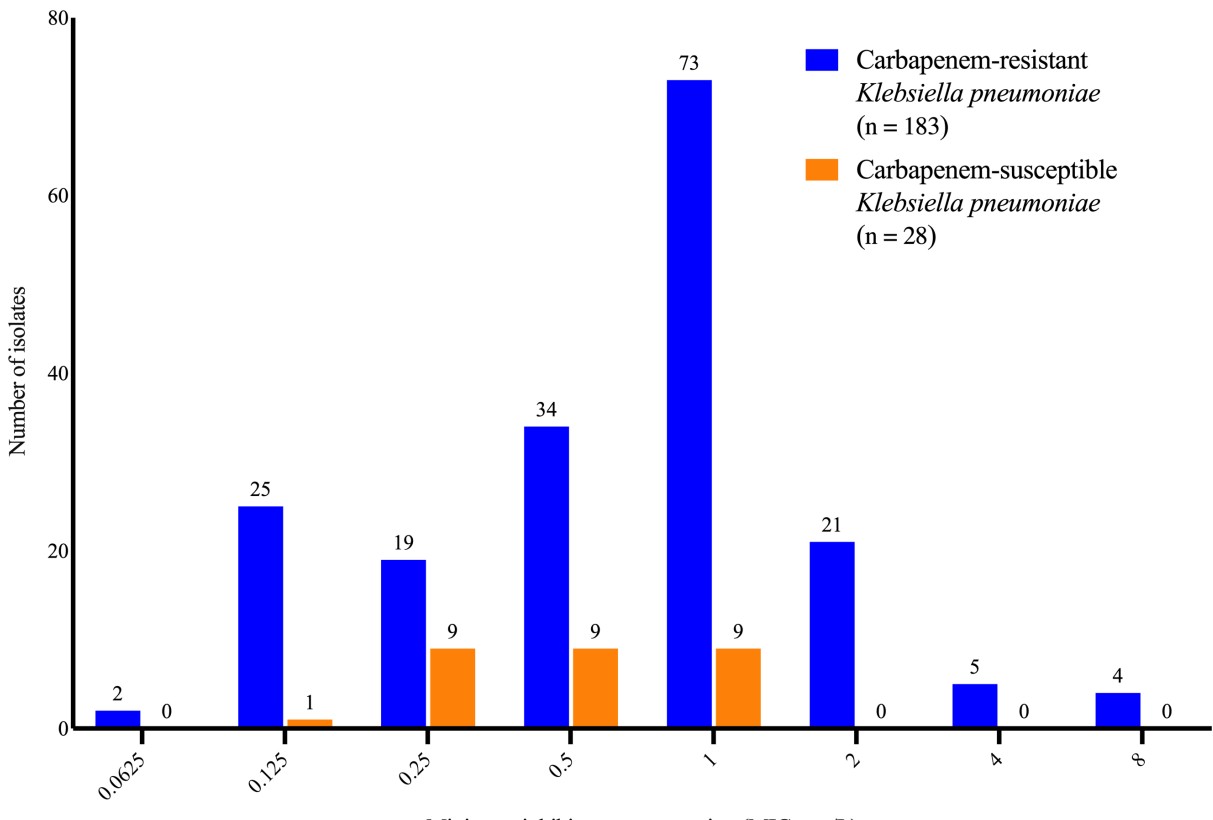

**FIG 1** The distribution of minimum inhibitory concentration (MIC) of eravacycline among 211 clinical *Klebsiella pneumoniae* isolates, comprising 183 carbapenem-resistant (CRKP, blue) and 28 carbapenem-susceptible (CSKP, orange) strains. Based on the proposed susceptibility breakpoint (≤1 mg/L), all CSKP isolates (100%) and 153 CRKP isolates (84%) were classified as susceptible to eravacycline. The MIC values are expressed in mg/L.

KP-515, and KP-487) when eravacycline was combined with polymyxin B (Table 5). However, only 3% and 7% of eravacycline-resistant CRKP strains exhibited a synergistic effect when eravacycline was combined with meropenem and fosfomycin, respectively, and eravacycline combined with amikacin had no synergistic effect at all (Table 6).

## DISCUSSION

Eravacycline demonstrated an overall susceptibility rate of 86% against the 211 *K. pneumoniae* strains used in this study, which is consistent with the rate of 86.7% reported for *K. pneumoniae* in a similar investigation of Gram-negative bacilli (12). Yun et al. observed an MIC90 of 2 mg/L against 20 CRKP strains, which is consistent with our results (13). Although the MIC of eravacycline against CRKP is moderately high compared with its activity against CSKP, targeted modifications at key molecular sites are likely to

**TABLE 1** The MIC values and susceptibility rates of eravacycline against carbapenemase-producing CRKP strains[a]

| Strains | MIC (mg/L) | | | | | | | | MIC 50/90 (mg/L) | Susceptibility (%) |
|---|---|---|---|---|---|---|---|---|---|---|
| | 0.0625 | 0.125 | 0.25 | 0.5 | 1 | 2 | 4 | 8 | | |
| CRKP (*n* = 183) | 2 | 25 | 19 | 34 | 73 | 21 | 5 | 4 | 1/2 | 84 |
| KPC-2 | 2 | 25 | 19 | 31 | 54 | 18 | 4 | 3 | 1/2 | 84 |
| Metallo-β-lactamases | 0 | 0 | 0 | 1 | 12 | 2 | 1 | 1 | 1/4 | 77 |
| KPC-2+metallo-β-lactamases | 0 | 0 | 0 | 2 | 7 | 1 | 0 | 0 | 1/1 | 90 |
| CSKP (*n* = 28) | 0 | 1 | 9 | 9 | 9 | 0 | 0 | 0 | 0.5/1 | 100 |
| *K. pneumoniae* (*n* = 211) | 2 | 26 | 28 | 43 | 82 | 21 | 5 | 4 | 0.5/2 | 86 |

[a]CRKP: carbapenem-resistant *Klebsiella pneumoniae*; KPC-2: *Klebsiella pneumoniae* carbapenemase-2; MIC: minimum inhibitory concentration; MIC50: minimum inhibitory concentration for 50% of the organisms; MIC90: minimum inhibitory concentration for 90% of the organisms; and CSKP: carbapenem-susceptible *Klebsiella pneumoniae*.

**TABLE 2** The MIC values and susceptibility rates of tigecycline and eravacycline against CRKP isolates[a]

| Drugs | MIC (mg/L) | | | Susceptibility (%) |
|---|---|---|---|---|
| | MIC50 | MIC90 | MIC range | |
| Tigecycline | 2 | 4 | ≤0.125–16 | 87 |
| Eravacycline | 1 | 2 | ≤0.0625–8 | 84 |

[a]MIC50: minimum inhibitory concentration for 50% of the organisms and MIC90: minimum inhibitory concentration for 90% of the organisms.

bolster its *in vitro* antibacterial performance, supporting its promise as a next-generation tetracycline for resistant infections. A prior investigation of the *in vitro* susceptibility of omadacycline revealed 100% resistance among class A carbapenemase producers, whereas susceptibility reached 82.3% in class B carbapenemase (metallo-β-lactamase)-producing isolates (14). In contrast to omadacycline, eravacycline exhibited no significant differences in *in vitro* susceptibility to CRKP isolates across distinct carbapenemase classes in this study. These comparative findings highlight eravacycline's structural stability against diverse carbapenem-hydrolyzing enzymes.

In addition, tigecycline had an 87% susceptibility rate among the 183 CRKP strains. Nonetheless, eravacycline displayed lower MIC50/90 values (1 and 2 mg/L) than tigecycline (2 and 4 mg/L), which is in agreement with a previous study in which eravacycline consistently exhibited more favorable MIC values (15). Furthermore, pharmacokinetic/pharmacodynamic (PK/PD) analyses suggest that eravacycline may offer a significant therapeutic advantage over tigecycline for the treatment of CRKP infections (16). These findings underscore the clinical potential of eravacycline and the need for further investigation into its efficacy, especially in combination regimens and across diverse patient populations.

In this study, 30 eravacycline-resistant strains were evaluated in combination with polymyxin B, meropenem, amikacin, and fosfomycin. The checkerboard assays indicated that six strains (20%) exhibited synergistic effects with polymyxin B. Previous research has found that eravacycline combined with polymyxin B is the most potent regimen against carbapenem-resistant *Escherichia coli* and *K. pneumoniae*, with a reported synergy rate of 30%–35% (13). A more recent study showed a 60% synergy rate against carbapenem-resistant *E. coli* (17). These findings highlight the potential of an eravacycline–polymyxin B combination as a treatment for CRKP infection. Eravacycline combined with certain β-lactam antibiotics (e.g., ceftazidime or imipenem) exhibited synergy rates of >50% when used to treat carbapenem-resistant *Acinetobacter baumannii* (CRAB) (13). However, our results did not show such a high proportion of synergistic effect when eravacycline was combined with β-lactam antibiotics, indicating that further research is required to clarify the conditions under which these combinations are most effective.

Previous research has illuminated the interactions that occur between cationically charged polymyxin B molecules and the anionic lipopolysaccharides found in the outer membrane of Gram-negative bacilli (18). These studies have demonstrated that the engagement between polymyxin B and the outer membrane facilitates an increased penetration of eravacycline, which in turn effectively breaches the bacterial membrane to inhibit protein synthesis, thereby amplifying its antibacterial performance (19).

**TABLE 3** The MIC values and susceptibility rates of tigecycline against carbapenemase-producing CRKP strains[a]

| Strains | MIC (mg/L) | | | | | | | | MIC50/90 (mg/L) | Susceptibility (%) |
|---|---|---|---|---|---|---|---|---|---|---|
| | 0.125 | 0.25 | 0.5 | 1 | 2 | 4 | 8 | 16 | | |
| CRKP (*n* = 183) | 3 | 6 | 23 | 37 | 91 | 15 | 5 | 3 | 2/4 | 87 |
| KPC-2 | 2 | 4 | 18 | 35 | 79 | 12 | 4 | 2 | 2/4 | 88 |
| Metallo-β-lactamases | 1 | 1 | 2 | 1 | 9 | 2 | 1 | 0 | 2/4 | 82 |
| KPC-2+ metallo-β-lactamases | 0 | 1 | 3 | 1 | 3 | 1 | 0 | 1 | 1/4 | 80 |

[a]CRKP: carbapenem-resistant *Klebsiella pneumoniae*; KPC-2: *Klebsiella pneumoniae* carbapenemase-2; MIC: minimum inhibitory concentration; MIC50: minimum inhibitory concentration for 50% of the organisms; and MIC90: minimum inhibitory concentration for 90% of the organisms.

**TABLE 4** The MIC values and susceptibility rates of eravacycline-resistant CRKP isolates[a]

| Drugs | MIC (mg/L) | | | |
|---|---|---|---|---|
| | MIC50 | MIC 90 | MIC range | Susceptibility (%) |
| Eravacycline | 2 | 4 | 2–8 | 0 |
| Meropenem | 128 | 256 | 64–256 | 0 |
| Polymyxin B | 0.125 | 2 | 0.125–16 | 90 |
| Amikacin | 2 | 2 | 2–8 | 4 |
| Fosfomycin | 256 | 256 | 64–256 | 0 |

[a]CRKP: carbapenem-resistant *Klebsiella pneumoniae*;MIC: minimum inhibitory concentration; MIC50: minimum inhibitory concentration for 50% of the organisms; and MIC90: minimum inhibitory concentration for 90% of the organisms.

Consequently, the combination of eravacycline and polymyxin B exhibits a synergistic effect against *E. coli* and *K. pneumoniae*, underscoring the potential of this therapeutic strategy to combat these pathogens.

However, excessive damage inflicted on the membranes by polymyxin B can provoke a stress response within the bacteria. This response can activate mechanisms for

**TABLE 5** The checkerboard assay for eravacycline-based combinations in eravacycline-resistant CRKP isolates[a]

| Strains | MIC (mg/L) | | | FICI |
|---|---|---|---|---|
| | ERV | PB | ERV/PB | |
| Synergistic effects | | | | |
| KP-487 | 2 | 0.5 | 0.25/0.125 | 0.38 |
| KP-515 | 2 | 4 | 0.5/0.5 | 0.38 |
| KP-569 | 8 | 16 | 1/2 | 0.25 |
| KP-572 | 8 | 2 | 0.25/0.5 | 0.28 |
| KP-530 | 2 | 8 | 0.5/0.25 | 0.28 |
| KP-638 | 2 | 2 | 0.5/0.125 | 0.31 |
| Indifferent effects | | | | |
| KP-465 | 2 | 0.125 | 0.125/0.125 | >1 |
| KP-492 | 2 | 0.125 | 0.125/0.125 | >1 |
| KP-494 | 4 | 0.25 | 2/0.125 | 1 |
| KP-500 | 2 | 0.25 | 1/0.25 | >1 |
| KP-501 | 2 | 0.25 | 0.125/0.25 | >1 |
| KP-507 | 2 | 0.125 | 0.5/0.125 | >1 |
| KP-516 | 8 | 0.125 | 0.5/0.125 | >1 |
| KP-519 | 2 | 0.125 | 1/0.125 | >1 |
| KP-521 | 2 | 1 | 1/1 | >1 |
| KP-526 | 2 | 0.125 | 0.25/0.125 | >1 |
| KP-527 | 2 | 0.125 | 0.5/0.125 | >1 |
| KP-558 | 2 | 1 | 1/0.5 | 1 |
| KP-561 | 2 | 1 | 0.5/1 | >1 |
| KP-573 | 8 | 0.125 | 0.125/0.125 | >1 |
| KP-578 | 2 | 0.125 | 1/0.125 | >1 |
| KP-579 | 2 | 0.25 | 1/0.25 | >1 |
| KP-580 | 4 | 1 | 2/0.5 | 1 |
| KP-624 | 4 | 0.125 | 0.25/0.25 | >2 |
| KP-626 | 2 | 0.125 | 0.25/0.25 | >2 |
| KP-630 | 4 | 0.125 | 0.125/0.25 | >2 |
| KP-637 | 2 | 0.25 | 1/0.125 | 1 |
| KP-656 | 4 | 0.125 | 0.5/0.125 | >1 |
| KP-662 | 2 | 0.125 | 0.5/0.125 | >1 |
| KP-668 | 2 | 0.125 | 0.125/0.125 | >1 |

[a]MIC: minimum inhibitory concentration; ERV: eravacycline; PB: polymyxin B; and FICI: fractional inhibitory concentration index.

**TABLE 6** The checkerboard assay for eravacycline-based combinations in eravacycline-resistant CRKP isolates

| Drug combinations | Synergistic effects (%) | Indifferent effects (%) |
|---|---|---|
| Eravacycline + meropenem | 3 | 97 |
| Eravacycline + amikacin | 0 | 100 |
| Eravacycline + fosfomycin | 7 | 93 |

membrane repair or lead to alterations in metabolism (20). These changes may decrease the accumulation of eravacycline within cells, thereby reducing its performance against microbial infections. This insight highlights the critical need for a thorough understanding of drug antimicrobial mechanisms when devising combination therapies for future clinical applications.

This study had several limitations. The checkerboard assay used to evaluate combination activity may not correlate well with *in vivo* studies. Further validation is required in pharmacokinetic/pharmacodynamic models and *in vivo* infection models.

## Conclusion

The overall *in vitro* susceptibility rate of eravacycline against *K. pneumoniae* strains was 86%. Even in the CRKP strains, it reached 84%, which was independent of the type of carbapenemases. The checkerboard assays found that a combination of eravacycline and polymyxin B had a synergistic effect on 20% of eravacycline-resistant CRKP strains. These findings indicate the substantial clinical potential of eravacycline as a treatment option for *K. pneumoniae* infections.

## MATERIALS AND METHODS

### Strain isolation

All *K. pneumoniae* strains analyzed in this study were isolated from body fluid specimens (including sputum, blood, and urine) collected from patients at Shanghai Tenth People's Hospital between January 2023 and December 2024. To ensure uniqueness, duplicate strains from the same patient's body fluid culture collection were removed. The characterization of these isolates was performed using the VITEK-MS automated microbiology system (bioMérieux, Marcy L'Etoile, France). The NG-Test CARBA5 Carbapenemase Assay Kit determined the carbapenemase type present in the CRKP strains following the manufacturer's instructions (Fosun Diagnostic Technology, China).

### *In vitro* susceptibility testing

The MICs of eravacycline, tigecycline, meropenem, amikacin, and fosfomycin against clinical isolates were determined using a broth microdilution method following the CLSI M07-A9 guidelines (21). Before susceptibility testing, all clinical isolates and the control strain, *E. coli* ATCC 25922, were subcultured twice on fresh Mueller-Hinton (MH) agar plates to ensure purity and viability. The MICs were interpreted according to the following criteria: eravacycline, China Antimicrobial Surveillance Network (ChinaCAST) guidelines (http://www.chicast.cn); tigecycline, thresholds recommended by the U.S. Food and Drug Administration (https://www.fda.gov/drugs/development-resources/tigecycline-injection-products); polymyxin B, EUCAST 2021 colistin susceptibility breakpoints (https://www.eucast.org/ast_of_bacteria/mic_determination); and other antibiotics, CLSI M100-Ed34 (2024) standards (https://clsi.org/).

### Molecular characterization

The carbapenemase resistance genes were screened using PCR amplification with specific primers. The following primers were used: *blaNDM* (F: 5′-GTAGTGCTCAGTGTCG GCAT-3′; R: 5′-GGGCAGTCGCTTCCAACGGT-3′), *blaKPC* (F: 5′-ATGTCACTGTATCGCCGTC-3′; R: 5′-TTTTCAGAGCCTTACTGCCC-3′), *blaOXA-48* (F: 5′-GATCGGATTGGAGAACCAGA-3′; R:

5′-ATTTCTGACCGCATTTCCAT-3′), *blaIMP* (F: 5′-GGAATAGAGTGGCTTAATTCTC-3′; R: 5′-CCA AACCACTACGTTATC-3′), and *blaVIM* (F: 5′-GTGTTTGGTCGCATATCGC-3′; R: 5′-CGCAGCACC AGGATAGAAG-3′).

## Checkerboard assays

Checkerboard assays were performed to evaluate the synergistic activity of eravacycline combined with meropenem, polymyxin B, fosfomycin, or amikacin. Each antibiotic was prepared in twofold serial dilutions (0.125–16 mg/L) and dispensed (25 µL/well) into 96-well microplates. Fresh bacterial suspensions were adjusted to a 0.5 McFarland standard using colonies harvested from Mueller-Hinton agar and diluted 1:100 in sterile saline. Aliquots (50 µL) of the diluted suspension were inoculated into each well and incubated at 35°C for 18–24 h under ambient air. The MIC for each antibiotic combination was determined post-incubation. The interactions between antibiotic combinations were assessed using the fractional inhibitory concentration index (FICI) as follows:

$$\text{FICI} = \frac{\text{MIC of durg A combination}}{\text{MIC of drug A alone}} + \frac{\text{MIC of drug B in combination}}{\text{MIC of drug B alone}}.$$

Interaction outcomes were categorized as synergism (FICI ≤ 0.5), additive (0.5<FICI≤1), indifference (1<FICI≤4), or antagonism (FICI > 4) based on standardized criteria for checkerboard assays.

## ACKNOWLEDGMENTS

We thank Everest Medicines Ltd. (Shanghai, China) for providing analytical grade eravacycline.

This study was supported by the Clinical Research Project of the Shanghai Municipal Health Commission (No. 202340254) and Shanghai Hospital Development Center Foundation (No. SHDC22023234).

S.W. and Y.D. conceived and designed the project. Y.G., Y.Z., X.J., and H.G. helped collect the *K. pneumoniae* strains. Y.L. and H.G. conducted the *in vitro* susceptibility testing and checkerboard test. Y.Z. and X.J. performed the data analysis. Y.G. and Y.D. wrote the manuscript. S.W. and Y.L. revised the manuscript. All authors contributed to the article and approved the submitted version.

## AUTHOR AFFILIATIONS

[1]Department of Critical Care Medicine, Shanghai Tenth People's Hospital, School of Medicine, Tongji University, Shanghai, China
[2]Intensive Care Medical Centre, Tongji Hospital, School of Medicine, Tongji University, Shanghai, China

## AUTHOR ORCIDs

Yingying Du  http://orcid.org/0000-0002-3945-8778
Sheng Wang  http://orcid.org/0000-0003-3439-4696

## FUNDING

| Funder | Grant(s) | Author(s) |
| --- | --- | --- |
| Cinical Research Project of the Shanghai Municipal Health Commission | No. 202340254 | Sheng Wang |
| Shanghai Hospital Development Center Foundation | No. SHDC22023234 | Sheng Wang |

## ETHICS APPROVAL

The clinical strains used in this study were obtained from individuals admitted to the Shanghai Tenth People's Hospital. These isolates were collected as part of the standard procedural measure when performing fluid cultures of infection sites in patients presenting with potential infections. The incorporation of these clinical isolates into *in vitro* susceptibility trials was approved by the Ethics Committee of Shanghai Tenth People's Hospital under the ethics approval code SHSY-IEC-4.1/18–74/01. The execution of this research was in strict alignment with the Declaration of Helsinki.

## ADDITIONAL FILES

The following material is available online.

### Open Peer Review

**PEER REVIEW HISTORY (review-history.pdf).** An accounting of the reviewer comments and feedback.

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
