## [Reviewer comments · Microbiology Spectrum]

Microbiology Spectrum

Antibacterial Activity of Eravacycline Against *Klebsiella pneumoniae* Isolates: An *in vitro* Study

Yuanzhi Gong, Yuhao Liu, Yunlou Zhu, Shikui Mu, Hanlu Gao, Xin Jing, Yingying Du, and Sheng Wang

Corresponding Author(s): Sheng Wang, Tongji Hospital Affiliated to Tongji University

Review Timeline:

Submission Date:	October 10, 2024
Editorial Decision:	February 2, 2025
Revision Received:	April 3, 2025
Accepted:	April 14, 2025

Editor: Andrea Prinzi

Reviewer(s): Disclosure of reviewer identity is with reference to reviewer comments included in decision letter(s). The following individuals involved in review of your submission have agreed to reveal their identity: Benno H. Ter Kuile (Reviewer #1); Mor N Lurie-Weinberger (Reviewer #2)

Transaction Report:

DOI: <https://doi.org/10.1128/spectrum.02564-24>

Re: Spectrum02564-24 (Antibacterial Activity of Eravacycline Against *Klebsiella pneumoniae* Isolates: An *in vitro* Study)

Dear Dr. Sheng Wang:

Thank you for the privilege of reviewing your work. Below you will find my comments, instructions from the Spectrum editorial office, and the reviewer comments.

Throughout the manuscript, the terms "efficacy" and "effectiveness" are used, and are often inappropriate. Effectiveness should not be used to describe the findings in this study, since the definition of effectiveness is the ability of an intervention to produce a desired effect in real-world settings, most often in actual patients. Efficacy represents the ability of an intervention to produce a desired effect under ideal conditions that are controlled. This is not an interventional study, and since the entire aim is to assess the performance of eravacycline (and other antibiotics), I would recommend that the authors use terms like "performance, susceptibility, resistance" instead.

Line 22- please change to Enterobacterales

Line 53- please include the manufacturer information in parentheses as you have done in the lines that follow

Line 60 - is this meant to be "isolates?"

Lines 60-61 - what does "incubated on MH agar plates twice" mean? Does this mean the isolates were sub-cultured twice prior to performing susceptibility testing? If so, please clarify.

Lines 61-63 - I suggest re-phrasing this to clarify that ChinaCAST was used to interpret the MICs for all antibiotics tested.

Lines 91-91 - as currently written, 183 strains were susceptible, which would be 100% of the CRKP isolates. This does not match the data described in Table 1 and elsewhere in the paper, please correct.

As noted by the reviewers, the tables and overall presentation of data are confusing and potentially inaccurate. These should be corrected (see reviewer comments) if the manuscript is to be accepted for publication.

There were concerns that the English language usage in the manuscript might make it difficult to properly evaluate the science. The ASM Journals webpage provides links to various language editing services (<https://journals.asm.org/writing-your-paper#language-editing-services>). You may consider using these services when revising your manuscript. The use of these services will have no direct bearing on the editorial decision. ASM has no affiliation with these companies.

Revision Guidelines

Publication Fees: For information on publication fees and which article types are subject to charges, visit our website. If your manuscript is accepted for publication and any fees apply, you will be contacted separately about payment during the production

process; please follow the instructions in that e-mail. Arrangements for payment must be made before your article is published.

Sincerely,
Andrea Prinzi
Editor
Microbiology Spectrum

Reviewer #1 (Comments for the Author):

Please see the attached file

Reviewer #2 (Comments for the Author):

Review of the paper: "Antibacterial Activity of Eravacycline Against *Klebsiella pneumoniae* Isolates: An in vitro Study"

The study focuses on 211 *K. pneumoniae* isolates that were selected for an in vitro susceptibility to the novel tetracycline antibiotic eravacycline. Utilizing the ChinaCAST breakpoints, eravacycline is described to have achieved an overall efficacy rate of 85.78%. This effectiveness rate against CRKP was 83.61% and a 100% efficiency against CSKP strains. Also, the researchers checked eravacycline combined with polymyxin B and show 24% of tested strains demonstrated a synergistic effect between the two antibiotics. They summarize that eravacycline has a substantial clinical potential in the management of *K. pneumoniae* infections.

Major Concerns

- Some editing would greatly improve the paper.
- I do not understand the percentages- 83% then 85% - it is not clear what you did there and why? Did you calculate the 211 then only 183? Then what is the 86% mean? "The MIC results show that tigecycline has an 86.89% susceptibility rate against CRKP"? This is unnecessarily confusing.
- The basic data of antibiotic susceptibility would be better presented first, before eravacycline evaluation.
- A large table summarizing all the resistance/ susceptibility data as a supplementary would have been a good way to allow readers to keep track of results.
- There are very few experiments described and very few results. This might be better suited to a short communication.

Minor Concerns

- I find the abbreviations strange. I have never seen *Klebsiella pneumoniae* called K.p. Also, CRKP and CSKP should be explained when they first appear, in the abstract.
- Out of the 211 isolates, 28 are sensitive to carbapenemases - would it not make sense to focus on the carbapenemase resistant strains as sensitive strains do not require a novel antibiotic?

Reviewer report Spectrum02564-24 Gong et al.,

This manuscript is a short and straightforward report on the *in vitro* effectiveness of the recently approved antibiotic eravacycline against *Klebsiella pneumoniae*. The science is solid and the message is of interest to the scientific community, but the usage of the English language is too often flawed. The manuscript needs to be thoroughly edited by an English language expert. Below a randomly chosen short paragraph is shown as an example. Even though it seems clear what the authors want to convey, the actual phrasing is incorrect.

Example of bad writing: Lines 44-47:

The investigation into the effectiveness of eravacycline against K.p has been marginally explored in previous research[12].However, the in vitro efficacy of eravacycline in combination with other antimicrobial agents against CRKP has not been thoroughly investigated. This study aims to evaluate the in vitro activity of eravacycline and potential combination therapy targeting CRKP.

The following would be more direct and lucid:

The effectiveness of eravacycline against *K.pneumoniae* has not yet been documented sufficiently [12]. Similarly, the efficacy against CRKP of eravacycline in combination with other antimicrobial agents has not been investigated. In this study the activity of eravacycline and the feasibility of combination therapies targeting CRKP are explored by testing over 220 strains *in vitro*.

Other major points:

The description of the data in table 1 is very confusing. The data are simple: out of 183 CRKP 30 are resistant to eravacycline (16.4%) and hence 83.6 % is susceptible. All CSPK tested are susceptible (100 %). That is not what is written in the text. In addition, the data would be easier to interpret if a bar graph was used instead of a table.

Table 4 is incoherent. I found no antibiotic by the name of “mempenem” in PubMed. I guess meropenem is meant. In that case, how can 100% of the CRKP be susceptible to meropenem?

Since in table 5 polymyxin B is used, it would be good to test the strains for resistance against this antimicrobial as well.

Tables 5 and 6 are not described in the Results section.

The discussion repeats too much the Results and gives a kind of an incoherent review. Instead, the authors should interpret their data in the framework of the literature and guide the reader to the conclusions.

Minor points:

The Materials and Methods section is underreferenced.

Line 84 “Result” = Results

The percentages are given in meaningless decimals, given that 211 samples were used in total. I suggest to round off to integers (e.g. 84% and not 83,97%)

The caption of table 5 is insufficient. The abbreviations must be defined.

Review of the paper: "Antibacterial Activity of Eravacycline Against *Klebsiella pneumoniae* Isolates: An in vitro Study"

Summary of Key Findings (200-250 words)

The study focuses on 211 *K. pneumoniae* isolates that were selected for an *in vitro* susceptibility to the novel tetracycline antibiotic eravacycline. Utilizing the ChinaCAST breakpoints, eravacycline is described to have achieved an overall efficacy rate of 85.78%. This effectiveness rate against CRKP was 83.61% and a 100% efficiency against CSKP strains. Also, the researchers checked eravacycline combined with polymyxin B and show 24% of tested strains demonstrated a synergistic effect between the two antibiotics. They summarize that eravacycline has a substantial clinical potential in the management of *K. pneumoniae* infections.

Major Concerns (at most 5-6):

- Some editing would greatly improve the paper.
- I do not understand the percentages- 83% then 85% - it is not clear what you did there and why? Did you calculate the 211 then only 183? Then what is the 86% mean? "The MIC results show that tigecycline has an 86.89% susceptibility rate against CRKP"? This is unnecessarily confusing.
- The basic data of antibiotic susceptibility would be better presented first, before eravacycline evaluation.
- A large table summarizing all the resistance/ susceptibility data as a supplementary would have been a good way to allow readers to keep track of results.
- There are very few experiments described and very few results. This might be better suited to a short communication.

Minor Concerns (at most 5-20 in bullet points):

- I find the abbreviations strange. I have never seen *Klebsiella pneumoniae* called *K.p.* Also, CRKP and CSKP should be explained when they first appear, in the abstract.
- Out of the 211 isolates, 28 are sensitive to carbapenemases – would it not make sense to focus on the carbapenemase resistant strains as sensitive strains do not require a novel antibiotic?

Confidential Comments to the Editor:

In general, editing would greatly improve the paper. There are all sort of strange mistakes, missing uppercase letters and strange phrasing of sentences.

Though there are many samples in this study, it appears that very few experiments had been done, so though the subject is interesting, the research itself is very lean and

concise, not the width and breadth I would expect from a research article. In fact this is a succession of antibiotic breakpoints... I believe that this is not on par with published SPECTRUM research papers. This could be a short communication, not a full article.

**Antibacterial Activity of Eravacycline Against *Klebsiella pneumoniae* Isolates: An
in vitro Study**

Yuanzhi Gong^{a#}; Yuhao Liu^{a#}; Yunlou Zhu^a; Hanlu Gao^a; Xin Jing^a; Yingying Du^{b*}; Sheng Wang^{b*}

^a Department of Critical Care Medicine, School of Medicine, Shanghai Tenth People's Hospital, Tongji University, Shanghai 200072, China

^b Intensive Care Medical Centre, Tongji Hospital, School of Medicine, Tongji University, Shanghai 200065, China

These authors contributed equally to this work.

*Corresponding authors: Yingying Du: 2211696@tongji.edu.cn, and Sheng Wang:

wangsheng@tongji.edu.cn

Review Editor**Point1**

This is not an interventional study, and since the entire aim is to assess the performance of eravacycline (and other antibiotics), I would recommend that the authors use terms like "performance, susceptibility, resistance" instead.

RESPONSE: Thank you for your thoughtful review and helpful suggestions. We fully agree with your recommendation regarding terminology. To better reflect the primary aim of this study, we replaced terms such as “effectiveness” and “intervention” with more appropriate terms like “performance,” “susceptibility,” and “resistance.” We have updated the relevant sections of the manuscript accordingly to ensure that the terminology aligns with the focus of the study.

Point2

Line 22- Please change to Enterobacterales

RESPONSE: Thank you for your careful review. In the revised manuscript (**page 3, line 38**), we have changed “Enterobacteriaceae” to “Enterobacterales.”

Point3

Line 53- please include the manufacturer information in parentheses as you have done in the lines that follow.

RESPONSE: Thank you for the suggestion. For the reagents mentioned, we have included the manufacturer information in parentheses (**pages 8, lines 186-190**).

Point4

Line 60 - is this meant to be "isolates?"

RESPONSE: Thank you for your comment. The term "isolates" refers to the isolated strains of *K. pneumoniae*.

Point5

Lines 60-61 - what does "incubated on MH agar plates twice" mean? Does this mean the isolates were sub-cultured twice prior to performing susceptibility testing? If so, please clarify.

RESPONSE: Thank you for raising this point. We have clarified this in the revised manuscript by rewording the sentence to explicitly describe this process (**page 9, lines 194-196**).

Point6

Lines 61-63 - I suggest re-phrasing this to clarify that ChinaCAST was used to interpret the MICs for all antibiotics tested.

RESPONSE: Thank you for your valuable feedback. The sections detailing antimicrobial susceptibility interpretation have been revised to clarify the breakpoints for various antibiotics explicitly (**page 9, lines 196-203**).

Point7

Lines 91-91 - as currently written, 183 strains were susceptible, which would be 100% of the CRKP isolates. This does not match the data described in Table 1 and elsewhere in the paper, please correct.

RESPONSE: Thank you for pointing out this inconsistency. We have revised the number of CRKP isolates in the revised manuscript (**page 4, lines 81-86 and Figure 1**).

Reviewer(s)' Comments to Author:

Reviewer 1

Comments to the Author

Point1

*This manuscript is a short and straightforward report on the in vitro effectiveness of the recently approved antibiotic eravacycline against *Klebsiella pneumoniae*. The science is solid and the message is of interest to the scientific community, but the usage of the English language is too often flawed. The manuscript needs to be thoroughly edited by an English language expert. Below a randomly chosen short paragraph is shown as an example. Even though it seems clear what the authors want to convey, the actual phrasing is incorrect.*

Example of bad writing: Lines 44-47:

*The investigation into the effectiveness of eravacycline against *K.p* has been marginally explored in previous research (12). However, the in vitro efficacy of eravacycline in combination with other antimicrobial agents against CRKP has not been thoroughly investigated. This study aims to evaluate the in vitro activity of eravacycline and potential combination therapy targeting CRKP.*

The following would be more direct and lucid:

*The effectiveness of eravacycline against *K. pneumoniae* has not yet been documented sufficiently (12). Similarly, the efficacy against CRKP of eravacycline in combination with other antimicrobial agents has not been investigated. In this study, the activity of eravacycline and the feasibility of combination therapies targeting CRKP are explored by testing over 220 strains in vitro.*

RESPONSE: Thank you for your suggestions. We have thoroughly revised the manuscript with the assistance of a professional English language editor. We have revised the examples you highlighted, along with other similar passages, to improve the overall readability (**page 4, lines 66-71**).

Point2

The description of the data in Table 1 is very confusing. The data are simple: out of 183 CRKP, 30 are resistant to eravacycline (16.4%), and hence 83.6 % are susceptible. All

CSKP tested are susceptible (100%). That is not what is written in the text. In addition, the data would be easier to interpret if a bar graph was used instead of a table.

RESPONSE: Thank you for your suggestions. Of the 183 CRKP isolates, 153 (84%) were found to be susceptible to eravacycline, whereas 16% were resistant. All tested CSKP isolates were susceptible (100%). The revised manuscript has replaced Table 1 by **Figure 1**.

Figure 1. The distribution of minimum inhibitory concentration (MIC) of eravacycline among 211 clinical *Klebsiella pneumoniae* isolates, comprising 183 carbapenem-resistant (CRKP, blue) and 28 carbapenem-susceptible (CSKP, orange) strains. Based on the proposed susceptibility breakpoint (≤ 1 mg/L), all CSKP isolates (100%) and 153 CRKP isolates (84%) were classified as susceptible to eravacycline. The MIC values are expressed in mg/L.

Point3

Table 4 is incoherent. I found no antibiotic by the name of "mempenem" in PubMed. I guess meropenem is meant. In that case, how can 100% of the CRKP be susceptible to meropenem?

RESPONSE: Thank you for pointing out this issue. The term "mempenem" appearing in the manuscript was a typographical error, and it has been corrected to "meropenem" in the revised manuscript (**Table 4**). All CRKP isolates in this study were resistant to meropenem. We have revised the manuscript and updated **Table 4**.

Point4

Since in Table 5, polymyxin B is used, it would be good to test the strains for resistance against this antimicrobial as well.

RESPONSE: Thank you for your suggestion. The susceptibility rates of CRKP to polymyxin B have been incorporated into the updated Table (**now presented as Table 4**).

Point5

Tables 5 and 6 are not described in the Results section.

RESPONSE: Thank you for bringing this to our attention. We have now updated the manuscript to properly described Tables 5 and 6 in the Results section (**page 5, lines 108–114**).

Point6

The Materials and Methods section is underreferenced.

RESPONSE: Thank you for your valuable feedback. We have carefully reviewed the manuscript and have ensured that all relevant methods, materials, and techniques are adequately referenced throughout (**pages 8-9, lines 191-203**).

Point7

Line 84 “Result” = Results

RESPONSE: Thank you for pointing out this typographical error. We have revised “Result” to “Results” on **line 72** in the revised manuscript.

Point8

The percentages are given in meaningless decimals, given that 211 samples were used in total. I suggest to round off to integers (e.g., 84% and not 83,97%)

RESPONSE: Thank you for your suggestion. We have revised the manuscript to round off the percentages to integers, as you recommended (**Tables 1, 2, and 3**).

Point9

The caption of Table 5 is insufficient. The abbreviations must be defined.

RESPONSE: Thank you for your comments. We have revised the caption to ensure that all abbreviations are clearly defined. (**Table 5**).

Reviewer 2

Comments to the Author

Point1

Some editing would greatly improve the paper.

RESPONSE: Thank you for your valuable feedback. The manuscript has undergone rigorous language editing and scientific refinement by professional native English-speaking editors.

Point2

I do not understand the percentages- 83% then 85% - it is not clear what you did there and why? Did you calculate the 211 then only 183? Then what is the 86% mean? "The MIC results show that tigecycline has an 86.89% susceptibility rate against CRKP"? This is unnecessarily confusing.

RESPONSE: Thank you for your comments. We evaluated the eravacycline susceptibility of 211 *K. pneumoniae* strains. The overall susceptibility rate for all strains was 86%, with a MIC range of 0.0625-8 mg/L. In the 183 CRKP strains, the susceptibility rate was 83.6% (153 CRKP strains were susceptible to eravacycline). The 86.89% susceptibility rate for tigecycline against CRKP was also calculated based on the 183 CRKP strains. The title and content of **Table 2** have been revised to accurately reflect the *in vitro* antimicrobial susceptibility results.

Point3

The basic data of antibiotic susceptibility would be better presented first, before eravacycline evaluation.

RESPONSE: Thank you for constructive feedback. The revised manuscript has been restructured to prioritize the presentation of comprehensive antibiotic susceptibility data, followed by a dedicated analysis of eravacycline's antimicrobial activity (**page 4, lines 81–86**).

Point4

A large table summarizing all the resistance/ susceptibility data as a supplementary would have been a good way to allow readers to keep track of results.

RESPONSE: Thank you for your helpful suggestion. We have added resistance/susceptibility data to Table 1 to enhance its accessibility and readability.

Table 1 The MIC values and susceptibility rates of eravacycline against carbapenemase-producing CRKP strains.

Strains	MIC (mg/L)								MIC 50/90 (mg/L)	Susceptibility (%)
	0.0625	0.125	0.25	0.5	1	2	4	8		
CRKP (n=183)	2	25	19	34	73	21	5	4	1/2	84
KPC-2	2	25	19	31	54	18	4	3	1/2	84
Metallo- β -lactamases	0	0	0	1	12	2	1	1	1/4	77
KPC-2+ Metallo-lactamases	0	0	0	2	7	1	0	0	1/1	90
CSKP (n=28)	0	1	9	9	9	0	0	0	0.5/1	100
K. pneumoniae (n=211)	2	26	28	43	82	21	5	4	0.5/2	86

CRKP: Carbapenem-resistant *Klebsiella pneumoniae*; KPC-2: *Klebsiella pneumoniae* carbapenemase-2; MIC: Minimum inhibitory concentration; MIC 50: Minimum inhibitory concentration for 50% of the organisms; MIC90: Minimum inhibitory concentration for 90% of the organisms; CSKP: Carbapenem-susceptible *Klebsiella pneumoniae*.

Point5

There are very few experiments described and very few results. This might be better suited to a short communication.

RESPONSE: Thank you for your valuable feedback. We understand your concern, but as we clearly stated in **Page 4, line 66-71**, this study aims to evaluate the *in vitro* susceptibility of eravacycline against *K. pneumoniae* using 211 clinical *K. pneumoniae* isolates and assessed combinatorial regimens to identify synergistic strategies against CRKP. The results are intended to fill an important gap in the current understanding of eravacycline's activity against CRKP, which remains underexplored. We believe this study contributes valuable insights into antimicrobial resistance, especially CRKP that is one of the most common nosocomial pathogens.

Point6

*I find the abbreviations strange. I have never seen *Klebsiella pneumoniae* called K.P. Also, CRKP and CSKP should be explained when they first appear in the abstract.*

RESPONSE: Thank you for pointing out these issues. In the revised manuscript, the species *Klebsiella pneumoniae* has been abbreviated as *K. pneumoniae*. Carbapenem-resistant *Klebsiella pneumoniae* (CRKP) and carbapenem-susceptible *Klebsiella pneumoniae* (CSKP) are now explained when they first appear (**Page 1, line 18-21; Page 3, line 39-41; Page 4, line 84**).

Point7

Out of the 211 isolates, 28 are sensitive to carbapenemases - would it not make sense to focus on the carbapenemase-resistant strains as sensitive strains do not require a novel antibiotic?

RESPONSE: You are absolutely right. This study mainly focused on the susceptibility of eravacycline against CRKP, CSKP was mentioned here only to better reflect the different sensitivity rates between CRKP and CSKP.

Re: Spectrum02564-24R1 (Antibacterial Activity of Eravacycline Against *Klebsiella pneumoniae* Isolates: An *in vitro* Study)

Dear Prof. Sheng Wang:

Thank you for making and integrating all reviewer edits, the revised manuscript is acceptable for publication. Please ensure that Figure 1 is appropriately labeled and has a caption.

Your manuscript has been accepted, and I am forwarding it to the ASM production staff for publication. Your paper will first be checked to make sure all elements meet the technical requirements. ASM staff will contact you if anything needs to be revised before copyediting and production can begin. Otherwise, you will be notified when your proofs are ready to be viewed.

Sincerely,
Andrea Prinzi
Editor
Microbiology Spectrum